# Desorption of Implanted Deuterium in Heavy Ion-Irradiated Zry-2

**Hideo Watanabe [1,*], Yoshiki Saita [2], Katsuhito Takahashi [2,3] and Kazufumi Yasunaga [4]**

[1] Research Institute for Applied Mechanics, Kyushu University, Kasuga-kouenn, Kasugashi, Fukuoka 816-8580, Japan

[2] Interdisciplinary Graduate School of Engineering Sciences, Kyushu University, Kasuga-kouenn, Kasugashi, Fukuoka 816-8580, Japan; koga@archkyushu-u.ac.jp (Y.S.); katsuhito.takahashi.hy@hitachi.com (K.T.)

[3] Hitachi, Co., Ltd., Omika, Hitachshi, Ibaraki 319-1292, Japan

[4] The Wakasa Wan Energy Research Center, Nagatani, Tsurugashi, Fukui 914-0192, Japan; kyasunaga@werc.or.jp

[*] Correspondence: watanabe@riam.kyushu-u.ac.jp

**Abstract:** To understand the degradation behavior of light water reactor (LWR) fuel-cladding tubes under neutron irradiation, a detailed mechanism of hydrogen pickup related to the point defect formation (i.e., a-component and c-component dislocation loops) and to the dissolution of precipitates must be elucidated. In this study, 3.2 MeV $Ni^{3+}$ ion irradiation was conducted on Zircaloy-2 samples at room temperature. Thermal desorption spectroscopy is used to evaluate the deuterium desorption with and without $Ni^{3+}$ ion irradiation. A conventional transmission electron microscope and a spherical aberration-corrected high-resolution analytical electron microscope are used to observe the microstructure. The experimental results indicate that radiation-induced dislocation loops and hydrides form in Zircaloy-2 and act as major trapping sites at lower (400–600 °C) and higher (700–900 °C)-temperature regions, respectively. These results show that the detailed microstructural changes related to the hydrogen pickup at the defect sinks formed by irradiation are necessary for the degradation of LWR fuel-cladding tubes during operation.

**Keywords:** light water reactor; zirconium alloys; nuclear fuel cladding; thermal desorption spectroscopy; transmission electron microscopy

## 1. Introduction

Hydrogen embrittlement of Zircaloy-2 is one of the main factors that limits the life of fuel rods in light water reactor (LWR) fuel-cladding tubes. Because the hydrogen centration exceeds the solid solubility of the metal, some hydrogen atoms are precipitated as hydride, causing hydride embrittlement. To understand how hydrogen pickup during operation influences the growth acceleration of the fuel rod, the effect of hydrogen on the neutron irradiation-induced microstructure should be studied. Hydrogen diffusivity under neuron irradiation is generally controlled by the trapping and de-trapping processes of hydrogen on the material. A dislocation loop is a radiation-induced defect cluster formed by the irradiation of neutrons [1,2], electrons [3,4], or charged particles [5–10]. Second-phase particles (SPPs) also form in Zircaloy-2 [11]. The size distribution of the SPPs and the chemical composition of the cladding tube affect the tube's corrosion rate inside boiling water reactors (BWRs) [12–17]. However, these SPPs are unstable during neutron irradiation and undergo an amorphous transformation resulting in the decomposition and redistribution of the constituent elements the precipitates into defect sinks [15–17]. Ion irradiation has been used on nuclear materials in multiple studies because, unlike neutron irradiation, the displacement per atom (dpa) level, irradiation temperature, and other irradiation conditions can be precisely controlled [5–10].

In this study, 3.2 MeV $Ni^{3+}$ ion irradiation is applied to Zircaloy-2 samples. The samples are injected with 5.0–30 keV $D_2^+$ ions to understand the trapping and de-trapping

process of hydrogen on $N^{3+}$ ion irradiated Zrycaloy-2. Thermal desorption spectroscopy (TDS) and conventional transmission electron microscopy (C-TEM) are conducted following the irradiation to evaluate the details of retention and desorption of the implanted deuterium and to identify the responsible traps.

## 2. Experimental Procedures

The Zircaloy-2 specimens were annealed at 630 °C for 2 h and subsequently air-cooled. Table 1 shows the results of the chemical analyses and the measurements of the hydrogen impurity levels of these specimens. The samples were irradiated at room temperature with 5.0–30 keV $D_2^+$ ions and 3.2 MeV $Ni^{3+}$ ions by an ion implanter and a tandem accelerator at Kyushu University. Figure 1a shows the ion irradiation chamber (with a duo-plasma ion gun) used for the $D_2^+$ irradiation, and Figure 1b shows the specimen holder used for the $D_2^+$ and $Ni^{3+}$ ion irradiation. Figure 2a shows the depth profile of the damage rate, and Figure 2b shows the concentration of $D_2^+$ ions irradiated at each accelerating voltage. The damage estimation was obtained from the Stopping and Range of Ions in Matter (SRIM) calculation [18], for which the threshold energy for displacement was assumed to be 40 eV. After irradiation at room temperature at a flux of $1.0 \times 10^{18}$ ions/m²s, the samples were transferred into the vacuum chamber of the TDS apparatus. After they were evacuated, the specimens were held at room temperature for a period of time (less than 2 h) and TDS measurements were conducted. During the heating, with a ramping rate of 1 °C/s up to 900 °C, the thermal desorption of HD (mass = 3) and $D_2^+$ (mass = 4) were measured using quadrupole mass spectroscopy. The desorption rate was calibrated by a He standard leak and corrected the relative ionization efficiency.

**Table 1.** Chemical composition of the materials used in the present study (wt%).

|  | Sn | Fe | Cr | Ni | H (wtppm) | Zr |
|---|---|---|---|---|---|---|
| **Zircaloy-2** | 1.38 | 0.15 | 0.09 | 0.05 | 46 | Bal. |

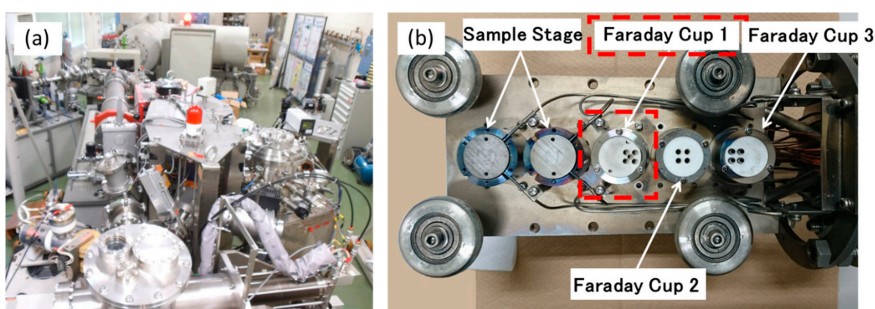

**Figure 1.** The triple ion beam facilities at RIAM Kyushu University: (**a**) tandem-type accelerator attached with two ion guns: terminal voltage 1.0 MeV and (**b**) specimen holder for ion irradiation.

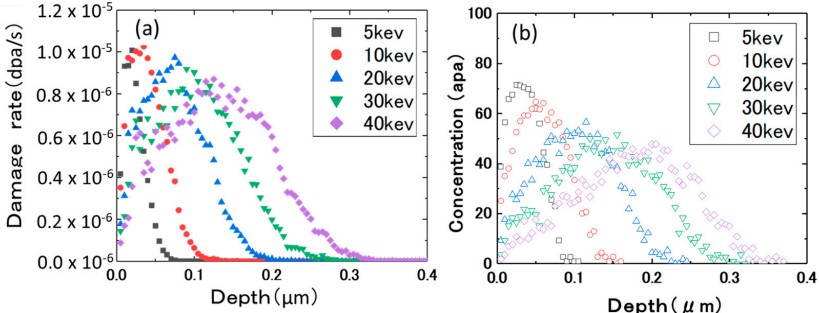

**Figure 2.** The depth profile of the damage rate (**a**) and the concentration of $D^+$ at $3.0 \times 10^{21}$ ions/m² (**b**) in the case of each accelerating voltage.

Figure 3 shows the damage distribution of the 3.2 MeV $Ni^{3+}$ ions and the impurity concentration ($Ni^{3+}$ ions) in the pure Zr following irradiation. The damage estimation was also obtained by the SRIM calculation. The samples for microscopy were thinned by the electropolishing method in which the electrolyte was a mixture of 50 mL perchloric acid and 950 mL acetic acid. During the thinning process, the electrolyte was held at $-40\ ^\circ$C and the potential applied was 25 V.

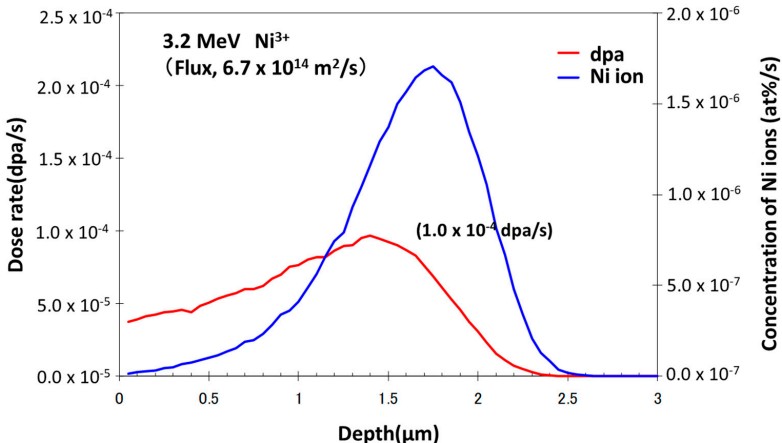

**Figure 3.** The SRIM calculation [18,19] of the damage distribution and the concentration of $Ni^{3+}$ atoms in Zr irradiated by $Ni^{3+}$ ions at 3.2 MeV. The values are estimated in the case of $6.7 \times 10^{14}$ ions/m²s.

The microstructure was observed before and after irradiation via C-TEM and using a spherical aberration (Cs)-corrected high-resolution analytical electron microscope (JEOL ARM200FC) operated at a voltage of 200 kV in a radiation-controlled area at Kyushu University.

## 3. Results and Discussions

### 3.1. Ion Dose and Energy Dependance of Thermal Desorption Behavior

Figure 4 shows the dose dependance of the thermal desorption spectrum after 5.0 keV $D_2^+$ ion irradiation at room temperature. Figure 4a–c show the cases of unirradiated samples, samples radiated with $3.0 \times 10^{21}$ ions/m², and samples irradiated with $10 \times 10^{21}$ ions/m², respectively. As shown in Figure 4a, the desorption stages of HD (mass = 3) and $D_2$ (mass = 4) were not detected prior to irradiation, but, after the $D_2^+$ ion irradiation, two major peaks at approximately 600 °C (mass = 3) and 800 °C (mass = 4) were detected. As demonstrated, the desorption stages were designated Peak A and Peak B on the lower temperature side. By increasing the irradiation dose from 3.0 to $10 \times 10^{21}$ ions/m², the total desorption of HD and $D_2$ increased from 1.8 to $5.5 \times 10^{19}$ ions/m², and from 2.5 to $6.6 \times 10^{18}$ ions/m², respectively. The increasing rate was 3.1 for HD and 3.8 for $D_2$. These values were nearly consistent with the increasing dose level values, particularly 3.3.

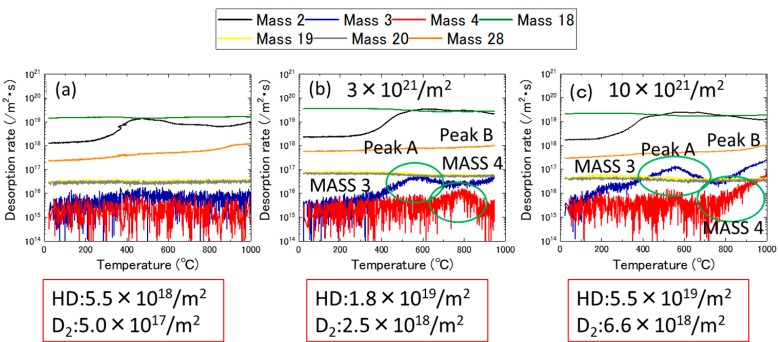

**Figure 4.** The dose dependance of the thermal desorption spectrum after 5.0 keV $D_2^+$ ion irradiation at room temperature: (**a**) before irradiation, (**b**) $3.0 \times 10^{21}$ ions/m², and (**c**) $10 \times 10^{21}$ ions/m².

Figure 5a,b shows the ion energy dependance of the thermal desorption spectrum after the administration of 5.0 keV $D_2^+$ ions and 30 keV $D_2^+$ ions with a dose of $3.0 \times 10^{21}$ ions/m$^2$, respectively. Figure 5b, shows that Peak B became prominent when the ion energy increased to 30 keV. In this figure, the total desorption of deuterium after 30 keV ion irradiation was much higher than that of the sample after 5.0 keV. Since the implanted deuterium atoms do not stay at the same position which is calculated in Figure 2, they diffuse to the thick region of the sample during irradiation and hydrides are formed. In the case of 5.0 keV irradiation, more deuterium atoms were released from the specimen surface than at 30 keV. Detailed estimation of deuterium atom diffusion during irradiation and desorption is needed.

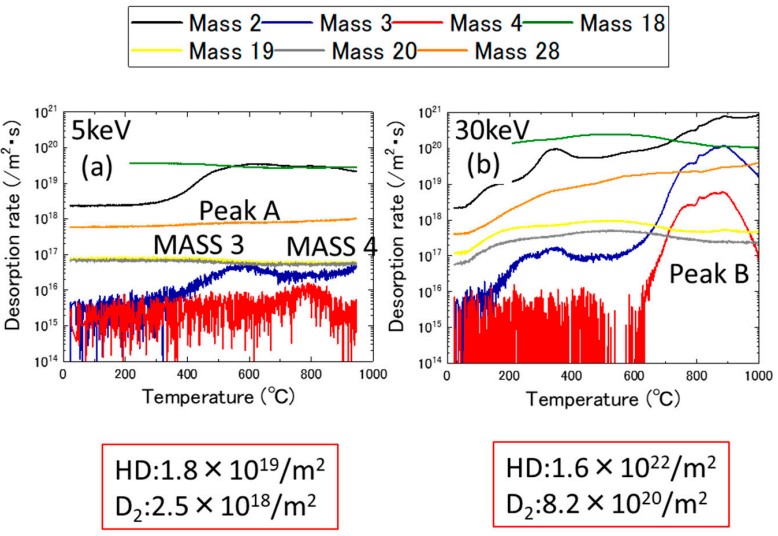

**Figure 5.** The energy dependance of the thermal desorption spectrum after the ion irradiation with a dose of $3.0 \times 10^{21}$ ions/m$^2$ at room temperature: (**a**) 5.0 keV and (**b**) 30 keV.

Figure 6a,b shows the microstructures of these samples. Hydrides were observed in the specimen's edge region following irradiation at 5.0 keV (Figure 6a). However, hydride formation was only detected in the thick region of the C-TEM samples following irradiation at 30 keV (Figure 6b).

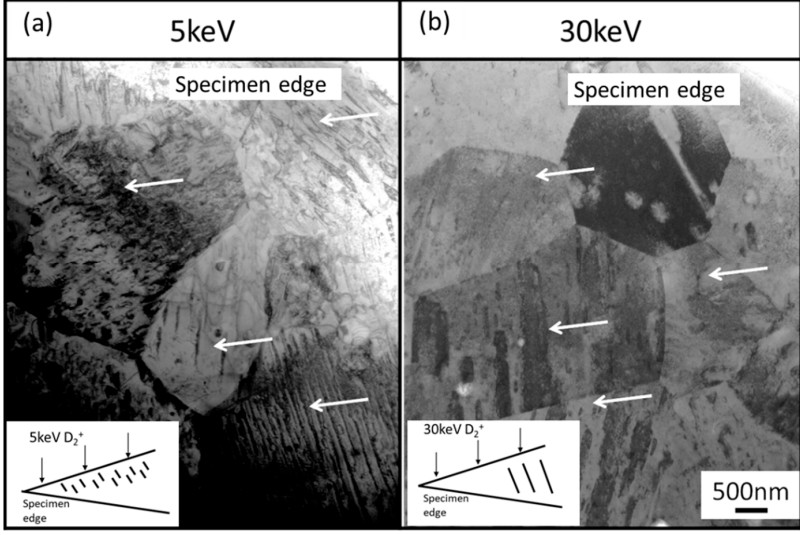

**Figure 6.** The energy dependance of the microstructure after the ion irradiation with a dose of $3.0 \times 10^{21}$ ions/m$^2$ at room temperature: (**a**) 5.0 keV and (**b**) 30 keV (the arrows show the hydrides formed by irradiation).

In our previous study of Zircaloy-2 [19], it was concluded that hydrides formed by annealing were stable up to 700 °C. The cross-sectional view of the sample showed that large hydrides were formed in the thick region of the samples. However, as Figure 6a shows, small hydrides formed in the specimen surface region by 5.0 keV $D_2^+$ ions irradiation were not stable in a higher temperature region (Peak B in Figure 5b). To investigate the thermal stability of these small hydrides in the surface region of the samples, the TEM samples was annealed at each temperature for 30 min, and the observation was conducted using a heating TEM holder. Figure 7 shows the thermal stability of these small hydrides, and they disappeared around 400 °C (in Peak A).

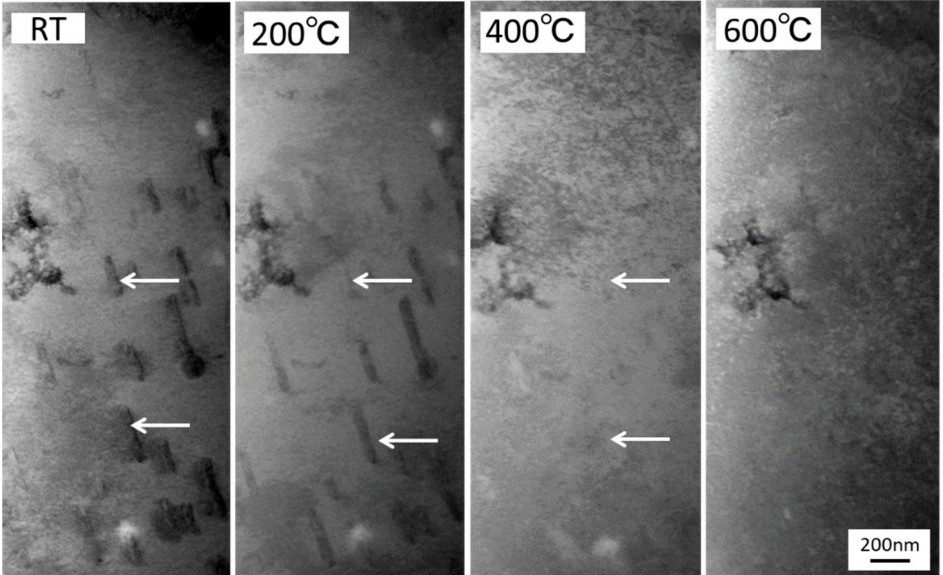

**Figure 7.** The thermal stability of the small hydrides formed in the surface region (30 keV, $3.0 \times 10^{21}$ ions/m$^2$ at room temperature). The hydrides (shown by arrows) disappeared at 400 °C.

### 3.2. Effects of Nickel Ion Irradiation

Figure 8a,b shows the microstructure after $Ni^{3+}$ ion irradiation at room temperature. After the irradiation (up to 3.0 dpa), high-density dislocation loops (approximately $4.1 \times 10^{20}$ m$^{-3}$) were detected. In the Zr alloys, interstitial-type dislocation loops (a-loops) and vacancy-type dislocation loops (c-loops) are known to form. Nakamichi et al. investigated the formation and growth process of a-loops in Zry-2 under electron irradiation using a high voltage electron irradiation (HVEM) [20]. Estimated migration energy for interstitial and vacancy were 0.17 eV and 1.0 eV, respectively. These dislocations formed in this study were identified as a-loops because of vacancy mobility at room temperature. A-loop formation was already saturated, and the loops were connected. Figure 9a,b shows the desorption spectrum before and after the 3.2 MeV $Ni^{3+}$ ion irradiation at room temperature, respectively. A new stage appeared in the position close to Peak A because of the $Ni^{3+}$ ion irradiation (up to 3.0 dpa).

### 3.3. Effects of Dislocations Formed by Cold Work

To determine the effects of the dislocations on the desorption spectrum, TDS experiments were conducted on cold-worked specimens in which dense dislocations were introduced. Figure 10a–d shows the desorption spectrum for the cold-worked specimens after irradiation with 5.0 keV $D^{2+}$ ion to the fluence of $3.0 \times 10^{21}$ ions/m$^2$ at room temperature. In Figure 10a,b, the desorption stage A shifted to the lower temperature side when the level of cold work increased. However, the desorption stage B remained consistent when the level of cold work increased. Table 2 summarizes the total amounts of HD and $D^2$. These values became saturated at 5% cold work and did not show any dependance on

the level of cold work. Table 3 summarizes the desorption temperature range and their responsible trapping site for deuterium obtained by the present study.

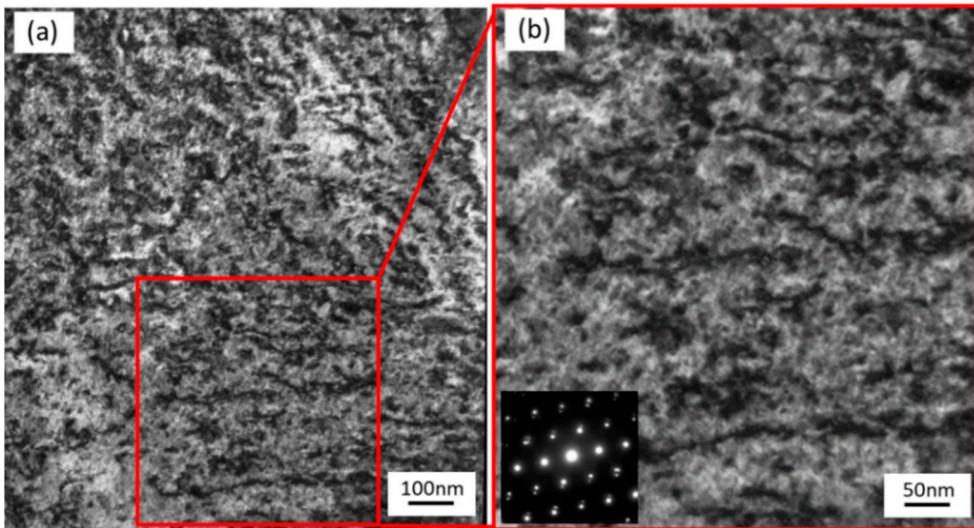

**Figure 8.** (**a**,**b**) The dislocation loops formed after 3.2 MeV Ni$^{3+}$ ion irradiation at room temperature. The irradiation dose was 3.0 dpa. Ion irradiation and TEM observation were conducted perpendicular to the <C> direction.

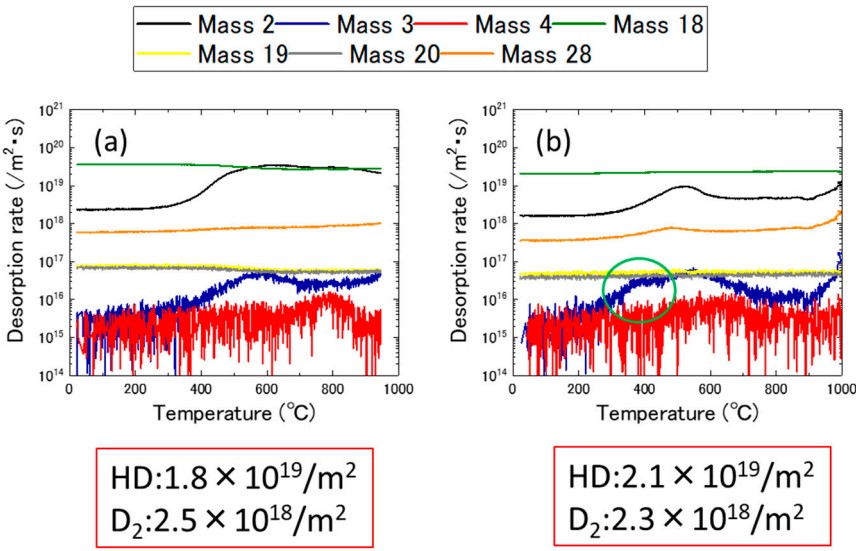

**Figure 9.** The thermal desorption spectra (**a**) before and (**b**) after 3.2 MeV Ni$^{3+}$ ion irradiation with a dose of 3.0 dpa at room temperature.

As was discussed in Section 3.2, Ni$^{3+}$ ion irradiation at room temperature induced a-loops into the samples. By increasing the level of cold work, stage A moved to the lower temperature side. Stage A corresponded to the recovery stage for weak trap sites as dislocation loops and hydrides formed in the surface region. Tangled dislocations (formed by cold work) and hydrides (formed in the thick region) contributed to stage B in the higher-temperature region.

In the Zircaloy-2 samples, a relatively large number (approximately 30 nm) of Zr$_2$(Fe,Ni) precipitates were formed in the matrix [19]. Small Zr(Fe,Cr)$_2$ precipitates were also found in the vicinity of the Zr$_2$(Fe,Ni) precipitates. The number density of the Zr(Fe,Cr)$_2$ precipitates was $2.0 \times 10^{19}$ m$^{-3}$ and that of the Zr$_2$(Fe,Ni) precipitates was $3.2 \times 10^{18}$ m$^{-3}$. The stability of these SPPs is necessary at higher dose levels because it is at these levels that the formation of c-loops and the dissolution of the SPPs are known to occur simultaneously [17,19].

Among these SPPs, the $Zr(Fe,Cr)_2$ precipitates are unstable during irradiation and undergo an amorphous transformation resulting in the decomposition and redistribution of other precipitates into defect sinks [15–17]. In this study, low-dose irradiation was chosen where phase stability of the SPPs was not essential. The irradiation dose was estimated to be approximately 18 dpa [21] at a burn-up of 50 GWd/t. This corresponded with the end of the life of the fuel rods for BWRs. The role of the redistributed precipitates and the number of c-loops formed at higher dose levels for hydrogen pickup are essential for the degradation of LWR fuel-cladding tubes during operation.

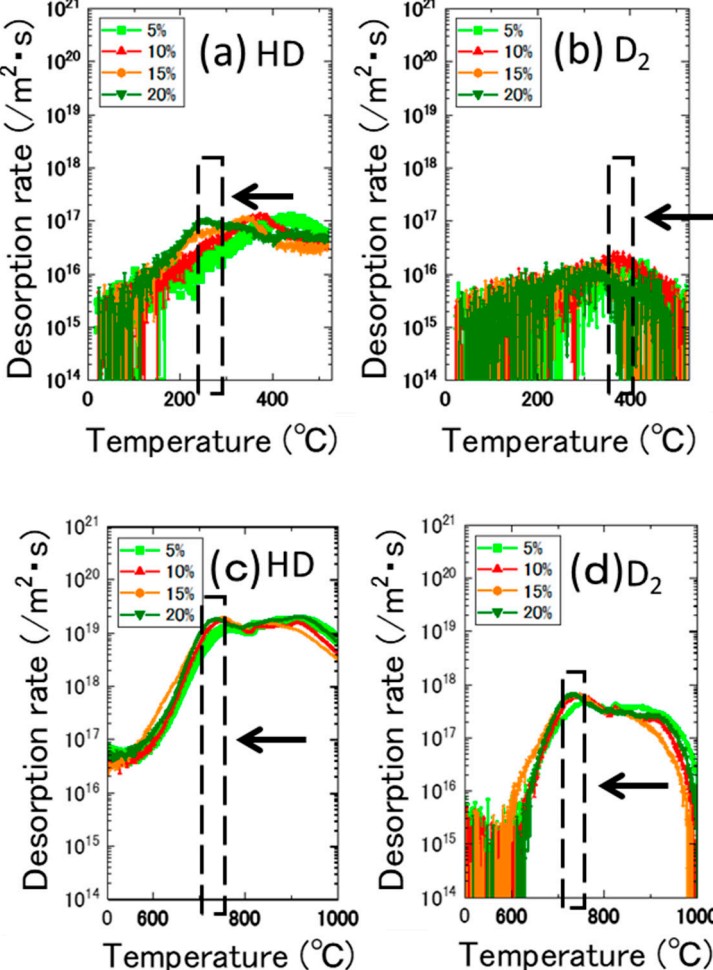

**Figure 10.** The amount of cold work dependance of the thermal desorption spectrum after 5.0 keV $D_2^+$ ion irradiation at room temperature: (**a**,**b**) lower temperature region and (**c**,**d**) higher-temperature region.

**Table 2.** Total desorption of HD and $D_2$ obtained in cold-worked samples.

| Cold Work (%) | HD ($m^{-2}$) | $D_2 (m^{-2})$ |
|---|---|---|
| 0 | $1.8 \times 10^{19}$ | $2.5 \times 10^{18}$ |
| 5 | $4.0 \times 10^{21}$ | $1.0 \times 10^{20}$ |
| 10 | $3.8 \times 10^{21}$ | $9.7 \times 10^{19}$ |
| 15 | $4.2 \times 10^{21}$ | $9.5 \times 10^{19}$ |
| 20 | $4.8 \times 10^{21}$ | $9.9 \times 10^{19}$ |
| 25 | $4.0 \times 10^{21}$ | $1.3 \times 10^{20}$ |

**Table 3.** Desorption temperature range and their responsible trapping site for deuterium.

| Stage | Temp. (°C) | Type of Trapping |
|---|---|---|
| Stage A | 400–600 | Weak Trap (dislocation loop, hydrides formed in surface region) |
| Stage B | 700–900 | Strong Trap (tangled dislocation, hydrides formed in thick region) |

## 4. Summary

In this study, the details of retention and desorption of implanted deuterium were investigated and the responsible traps in irradiated Zircaloy-2 were identified. The thermal desorption of deuterium was conducted on $Ni^{3+}$ ion-irradiated samples and cold-worked samples. The following conclusions have been drawn:

(1) The desorption spectrum shows two major desorption stages (named Peak A and Peak B) in the temperature ranges of 400–600 °C and 700–900 °C, respectively.

(2) As the thermal annealing experiments of ion-irradiated samples and cold-worked samples demonstrated, stage A corresponded to the recovery stage of weak trapping sites (i.e., the dislocation loops formed by irradiation and the hydrides that formed in the surface region of the specimens).

(3) Stage B corresponds to the strong trapping sites explained by the tangled dislocations and hydrides that formed in the thick region of the specimens.

**Author Contributions:** Conceptualization, H.W.; TDS, Y.S. and K.T.; TEM, K.Y.; project administration, H.W.; funding acquisition, H.W., All authors have read and agreed to the published version of the manuscript.

**Funding:** This research received no external funding.

**Acknowledgments:** This work was supported by the Collaborative Research Program of Research Institute for Applied Mechanics, Kyushu University.

**Conflicts of Interest:** The authors declare no conflict of interest.

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
