# Peer review of "Desorption of Implanted Deuterium in Heavy Ion-Irradiated Zry-2"

_qubs, doi:10.3390/qubs5020009_

Round 1
Reviewer 1 Report
The paper examines the strength of trapping sites for hydrogen in Zircaloy-2, using thermal desorption spectroscopy and TEM. Deuterium irradiations are used for implantation and Ni ion and cold working are used to generate defects. The paper concludes that there are two main types of hydrogen trapping site, weak traps (e.g. near surface regions), and strong traps (e.g. tangled dislocations). The paper acknowledges that the irradiation dose the material received is not near its end of life dose. This is justified as the paper only seeks to examine the effect of defect sinks without the additional complication of precipitate dissolution known to occur at higher irradiation doses.
There are a couple of sentences where the English is clunky, but on the whole the paper is readable and understandable. More detail is needed in several places, both to make it easier/quicker for a reader to understand and to justify the points being made. Figures 5, 7 and 8 in particular would benefit from further detail (see below).
Introduction: More emphasis could be put on the nature of defect trapping, which is the main thrust of the paper, time is given to discussing SPPs, which are at most tangentially related to the paper.
Line 21: Typo: ‘detail’ – ‘detailed’
Figure 1b: Typo: ‘Frady cup’ – ‘Faraday cup’
Figure 2: It should be stated how the damage and implantation profiles were calculated. SRIM is listed as the software used to calculate figure 3, but it would be better to explicitly state if it was used to produce figure 2 as well.
Methods: The irradiation temperature(s) should be given here, as well as further details pertaining to Figure 7.
Figure 6: The edge of the specimen appears to be at an angle, it would be useful to indicate the direction from which it was irradiated.
Line 107: What is meant by ‘thick regions’? Do they mean:
- The hydride density was so low that they were only seen in thicker TEM specimens-which have a larger volume of material, or
- They were only seen in regions further away from the surface?
If 2, then how are near surface hydrides shown in figure7?
Figure 5: The same number density of ions have been implanted to the material in (a) and (b), the only difference is the energy of the implanted deuterium ions. However, much more hydrogen/deuterium is desorbed after implanting the higher energy 30 keV deuterium. This is not discussed in any great detail and should be. Looking at the damage profiles in figure 2, the amount of damage induced by the implantation of both 5 and 30 keV ions is not too dissimilar, the damage profile is only wider for the 30 keV ions. How do the authors explain these results? Specifically, how do they relate them to the differences in trapping sites between the 5 and 30 keV deuterium ion irradiated materials.
Line 119: Reference figure that peak B is referring to
Line 120: Poor English – reword
Figure 7: How were the heat treatments performed? The regions shown appear to be the same, this would lead me to assume it was an in-situ anneal, but it is not stated. If an in-situ anneal was used, then the authors should address the issues around this. The point they are making is that near surface hydrides are trapped more weakly and so desorb at lower temperatures. But if this was done in-situ with an electron-transparent foil, then everything is near surface. Did the authors see the hydrides that were further from the irradiated surface also disappear? – In a thin foil they would also then be near surface.
Line 130-131: ‘These dislocations were identified as a-loops because of their size, shape, and vacancy mobility at room temperature’. What are they? Reference(s) are needed that detail the characteristic morphologies of a and c type loops.
Figure 8: A higher magnification image would be useful to show that the loops are connected here. The single image here is having to do a lot of heavy lifting for the statements made in the paragraph above.
Figure 10: It should be made clear what the arrows and boxes in the figure are referring to, either in the figure or in the figure description.
Line 164: TEM image(s) showing this would be nice.
Line 169: Reference(s) needed for the statement: ‘the formation of c-loops and the dissolution of the SPPs are known to occur simultaneously’.
Lines 173-4: How was the dose of 20 dpa estimated? Did the authors do it? If so, how? If not, reference needed.
Author Response
Thank you for your comments. Please see the attachment. Watanabe

Reviewer 2 Report
Report and suggestions on paper qubs-1172500 by Saita et al.
In the paper some results are presented, related to the understanding of degradation mechanisms in Zircaloy-2 under irradiation. The particular technique used involves deuterium implantation prior to heavy ion irradiation proper. Some of the Authors have already published on this subject, and the work under consideration presents some new results. Overall, the quality of the paper is reasonably good and use of post-irradiation desorption techniques in the study of Hydrogen de-trapping is interesting; experimental methodology looks sound and results presented in a rather consistent way. However, the manuscript needs some substantial editing; it is my opinion that some major changes are to be made in order to improve legibility. A brief list of suggestions follows.
- Figures 1 and 3 already appeared in a previous paper, and should be removed modifying the text accordingly; actually, Fig.3 of this work looks exactly the same as Fig. 1a of the #19 paper on the reference list. Old data should not be re-used, but simply recalled by using proper referencing. By the way, there seems to be a misprint on Fig.1 b; "Frady" cups are probably "Faraday" cups.
- In Fig.2, how were depth profile data obtained ? Probably by means of the same standard software used for the data of Fig 3. However, there is no clear reference anywhere and I think that a few words in the text would be in order.
- data related to thermal desorption (Figures 4,5,9,10); figures are too small, and it is difficult to follow the discussion (even on the PDF file). Enlarging these figures would contribute to the quality of the paper; possibly, some of them could be eliminated using some rephrasing in the text; however, I leave this last choice to the Authors.
Once these changes have been made, the paper could be considered for publication.
Author Response
Thank you for your comment. Please see the attachment.

Round 2
Reviewer 1 Report
I thank the authors for adressing my comments. I would have liked the introduction to focus a little more on the discussion of hydrides and trapping as this is the main investigation of the paper, although this is not strictly necessary for publication.
Apart from that, there are a couple of English corrections:
Line 43: Typo: ‘trapping’ not ‘tapping’
Text inserted in lines 108-114: English corrections:
In this figure, the total desorption of deuterium after 30keV ion irradiation was much higher than that of the sample after 5.0 keV ion irradiation. Since the implanted deuterium atoms does not stayed at the same position which is calculated in Figure 2. They diffuse to the thick region of the sample during irradiation and hydrides are formed. In the case of 5.0 keV irradiation, much more deuterium atoms were released from the specimen surface than the 30 keV irradiation. Detailed estimation of deuterium atom diffusion during irradiation and also desorption is needed.
Author Response
Please see the attachment. Watanabe

Author Response
Thank you. Please see attachment.
